# Rodent Model of Gender-Affirming Hormone Therapies as Specific Tool for Identifying Susceptibility and Vulnerability of Transgender People and Future Applications for Risk Assessment

**DOI:** 10.3390/ijerph182312640

**Published:** 2021-11-30

**Authors:** Roberta Tassinari, Francesca Maranghi

**Affiliations:** Center for Gender-Specific Medicine, Istituto Superiore di Sanità, 00161 Rome, Italy; francesca.maranghi@iss.it

**Keywords:** transgender, in vivo study, gender dysphoria

## Abstract

Transgenders (TGs) are individuals with gender identity and behaviour different from the social norms; they often undergo gender-affirming hormone therapy (HT). HT for TG men involves testosterone treatment and, for TG women, oestrogen plus androgen-lowering agents. Due—but not limited—to the lifelong lasting HT, usually TG people experience several physical and behavioural conditions leading to different and specific susceptibility and vulnerability in comparison to general population, including the response to chemical contaminants present in daily life. In particular, the exposure to the widespread endocrine disrupters (EDs) may affect hormonal and metabolic processes, leading to tissue and organ damage. Since the endocrine system of TG people is overstimulated by HT and, often, the targets overlap with ED, it is reasonable to hypothesize that TG health deserves special attention. At present, no specific tools are available to study the toxicological effects of environmental contaminants, including EDs, and the potential long-term consequences of HT on TG people. In this context, the development of adequate and innovative animal models to mimic gender-affirming HT have a high priority, since they can provide robust data for hazard identification in TG women and men, leading to more reliable risk assessment.

## 1. Introduction

The word ‘transgender’ (TG) is used to describe individuals whose gender identity and behaviour not completely and/or permanently match the sex assigned at birth; they include also gender-nonconforming people who may have a binary or nonbinary gender identity: binary, if they identify themselves as women (assigned male at birth) or men (assigned female at birth), whereas nonbinary individuals reject to be considered exclusively in masculine or feminine gender [1,2]. As a common feature, in TG people, gender identity, role or expression differ from what is considered normal for their assigned sex at birth in a given culture and historical period (https://www.infotrans.it/ (accessed on 20 October 2021). In this context, TG is an ‘umbrella’ term although terminology is continuously evolving and terms used in the past may become outdated and even perceived as pejorative [3]. Gender dysphoria is the distress caused by the discrepancy between a person’s assigned sex at birth and gender identity [4]. The prevalence rates of TG people are based on population studies and on the available data derived from people attending TG health clinics. In 2015, a meta-analysis in western countries (mainly Europe and the United States) found an overall prevalence for transsexualism (this is the diagnosis and term used in the papers) of 4.6 in 100,000 individuals, 6.8 for TG women (birth-assigned male, female gender identity) and 2.6 for TG men (birth-assigned female, male gender identity), with an increase in the reported prevalence during the last 50 years [5]. However, recent findings showed that the estimated proportion of gender-diverse individuals (those who are not cisgender, people whose sense of personal identity and gender corresponds with their birth sex) ranges between 0.1 and 3% of the population, depending on the inclusion criteria and where the studies were held [6,7,8]. It was found that TG identities have increased in recent years and the actual number of TG people may be higher because of the surrounding stigma and limited data collection. Moreover, it is interesting to note that TG prevalence among adults is at its highest in younger age groups (18–24 ≥ 25–64 ≥ 65 and older) [9]. The increase of TG prevalence in younger individuals could be attributed to a more inclusive socio-cultural context, which facilitates people to express themselves and to realize their gender identity objectives, in comparison to the socio-cultural context of older generations [1].

In the last years, due to the widespread presence of chemical contaminants in the environment, several authors hypothesized an association between exposure and increased gender dysphoria, in particular for endocrine disrupters (EDs). Such chemicals, both of natural and man-made origin, may mimic or interfere with the endocrine homeostasis and the exposure to EDs is associated with a plethora of effects, including developmental, reproductive, brain, immune, metabolic and other effects [10]. EDs can compete with natural ligands and alter the biochemical pathways of the brain and the development of secondary sex characteristics; this, in turn, may lead to modifications in normal behaviour or in gender development consistent with the assigned sex without a direct dysregulation of tissue differentiation [11,12,13]. Interestingly, a recent paper of Ramirez K et al. showed differential methylation of genes involved in brain neurodevelopment, gene expression and neuronal migration in TG people in comparison to cisgender population, allowing them to postulate the presence of specific methylation profiles in TG people [14]. Since chemical exposure during critical phases of peri- and post-natal development, but also in later stages, can induce DNA methylation, the impact of environmental factors on TG people should be carefully studied.

Not every TG person—either TG women or TG men—suffers from gender dysphoria and the need of medical intervention among TG people may vary; for some of them, social changes can be sufficient without further physical intervention, but many others have access to TG health services for gender-affirming treatments in the form of hormone therapy (HT) and/or through gender-affirming surgery. Gender-affirming treatments have shown to substantially reduce mental health problems, such as depression, anxiety and self-harm that TG people experience more than cisgender people do [15]. The HT is the least invasive intervention for gender affirming and it allows the individual to reach the desired physical characteristics; indeed, HT can be considered as a chronic treatment with drugs which side effects should not be neglected. Moreover, TG people undergo several physical and behavioural conditions—beyond HT—that may lead to different and specific susceptibility and vulnerability in comparison to the general population, including the response to chemical contaminants commonly present in daily life. In this respect, TG people may represent a susceptible sub-population group needing specific attention in the frame of risk assessment (RA), the scientific process aiming to identify and estimate the risks resulting from chemical or physical agents through all the possible routes of exposure [16]. For the cis-population as well as the subgroup of TG people, the RA is composed of four steps: (1) hazard identification—the qualitative aspect that sets the inherent properties of a substance or mixture of substances able to cause adverse effects in the organism, depending on the degree of exposure; (2) dose-response relation—the quantitative aspect defining the relationship between the dose and either the severity or the frequency of the effect(s); (3) exposure assessment—identification of the intensity and the duration or frequency of exposure to an agent; (4) risk characterization—quantification of the risk based on the above data [17].

In the frame of RA, particularly relevant considering TG people is the widespread exposure to EDs. In fact, they may affect different hormonal and metabolic processes, as the secretion, activation, synthesis, release and binding of hormones, and the exposure to EDs, even at low doses, can lead to tissue and organ damage [18]. Interestingly, a commonly observed phenomenon linked to ED exposure is the occurrence of sex-specific effects either in laboratory animals or in humans [19]. Moreover, EDs are present in many everyday products, including some plastic bottles and containers, liners of metal food cans, detergents, flame retardants, food, toys, cosmetics and pesticides, leading to a continuous exposure through the whole life [20]. Since the endocrine system of TG people using HT is overstimulated and they are—at the same time—exposed to contaminants targeting the endocrine system, it is reasonable to hypothesize that the health of TG people deserves special attention. At present, no specific tools are available to study the toxicological effects of EDs and other environmental contaminants, as well as the potential long-term consequences of HT on TG people. In this context, the development of adequate and innovative animal models to mimic gender-affirming HT therapies has a high priority. The present work aims to summarize the state of art on the main health problems encountered by TG people approaching HT, the available studies mimicking gender-affirming HT therapy and the scientific basis for the development and improvement of appropriate rodent models for the evaluation of the long-lasting and toxicological effects of gender-affirming HT, as a basis to drive a targeted RA for TG people.

## 2. Gender-Affirming Hormone Therapy

The major objectives of HTs are the replacement of endogenous sex hormone levels within the normal range for the affirmed gender (cisgender) and the regression of secondary sex characteristics of the designated gender. Potentially, HT should be administered lifelong to maintain the achieved masculinization or feminization, because, in general, HT has also proved to reduce morbidity and to improve TG people’s quality of life [15,21].

Considering TG men, masculinizing HT involves exogenous testosterone (T) administration. Key issues of gender-affirming T therapy include maintaining T levels in the cisgender male range (320–1000 ng/dL) [21] and controlling adverse effects. Different T formulations may be available and the most common route of administration is intramuscular (im) or subcutaneous (sc) injection of T enanthate or cypionate. More recently, transdermal administration (patches or gel) of longer-action T undecanoate showed to be effective, less invasive and suitable for long-term use. Oral administrations of T are infrequent because they require regular dose adjustment and the serum levels are often unpredictable. Other hormone medications, such as pregestational agents or gonadotropin-releasing hormone agonist (GnRH-a) are occasionally used, only if menstrual bleeding does not stop after a 1–12-month period of T administration [15,21]. The main effects of gender-affirming T therapy are: cessation of menses, body fat redistribution, clitoral enlargement, voice deepening and increased body hair in a more masculine distribution. Some masculinization effects can be seen as early as 1–2 months after T therapy initiation. Other sings of virilization, such as facial and/or body hair growth, change in fat distribution and increased muscle mass, take longer to fully develop—up to 5 years [15,21].

For TG women, feminizing HT usually includes oestrogen (oestradiol, E2), alone or in combination with androgen-lowering therapies. Indeed, the therapies used in different countries can consistently vary and a common protocol has not yet been established [15].

E2 is considered as first-line drug for TG women and E2 is administered transdermally, orally, or via injection. The dose level and the time of administration change based on the means of E2 administration. In most European countries, im injections are not available, as they can produce peaks in E2 lasting for about 2 weeks and reaching values up to six times higher than the E2 concentration in cisgender women. Such high serum E2 levels may potentially be associated with thromboembolic events. The oral use of E2 may be limited by the lower concentrations and fluctuating E2 levels due to first-pass metabolism into oestrone and oestrogen conjugates. Instead, transdermal E2 gives a more constant, non-fluctuating E2 serum level without metabolites, which may be advantageous in terms of cardio-metabolic side effects [22].

Anti-androgen treatment may include cyproterone acetate (CPA), GnRH-a or spironolactone. The CPA is used in Europe (50 mg daily) and its anti-androgenic effects are mainly caused by competitively blocking the androgen receptor (AR). It also influences sexual desire and spermatogenesis and showed anti-gonadotropic effect causing a decrease in luteinizing hormone (LH) and follicle-stimulating hormone (FSH) secretion. As CPA is a progestogen, it also has progestogen-like effects, such as changes in lipid metabolism [23]. The T-lowering agent most commonly used in the United States is spironolactone. Spironolactone is an antagonist of aldosterone and a moderate antagonist of the AR leading to a reduction in T levels. T concentrations usually do not decline up to cisgender ranges and only the use of spironolactone in combination with high doses of E2 (i.e., median dose of 6 mg/day) causes T decline in the cisgender range. However, no data are available on the adverse effects caused by E2 high doses in combination with spironolactone; nevertheless, differently from CPA, spironolactone has not been associated with increases in prolactin levels [23].

GnRH-a initially increases the concentrations of LH and FSH. However, over time, the overstimulation of the hypothalamic–pituitary–gonadal axis produces a decrease in LH and FSH levels and, subsequently, reduces T concentrations. Different from CPA and spironolactone, GnRH-a is administered by im or sc injection and the data on long-term safety are limited [23]. While oestrogens are considered a lifelong therapy, anti-androgens are generally discontinued after genital surgery (orchiectomy) or if oestrogen-only treatment has proven to be sufficient [22].

The principal physical changes in TG women are the development of breast tissue and the reduction in testicular volume. Sperm production declines and feminization by HT is associated with increased body fat and decrease in lean body mass, reduction in facial hair and improved skin softness [15].

HT can be essential to achieve well-being in TG people. However, to date, little is known regarding the health of TG men and women and the gaps in the knowledge are large. Medical and mental health outcomes, along with co-morbidity conditions over the life span, have not been completely ascertained nor have the specific consequences and the potential adverse effects of gender-affirming interventions or the impact on sexual function and fertility [24].

## 3. Specific Susceptibility and Vulnerability of Transgender People

From the standpoint of toxicological RA, TG people present several unique features linked to exposure scenarios and general vulnerability (Figure 1). Indeed, given the same levels of xenobiotics—above all EDs—in the living environment, internal exposure may be different due to the lifelong HTs, which might substantially alter TG people’s physiological status as well as modifying the expression of enzymes involved in xenobiotic metabolism. Specific examples are listed below.

### 3.1. Cardiovascular System

Cardiovascular diseases are one of the major causes of death worldwide for both men and women. Adult women usually have a lower incidence of cardiovascular diseases but a higher rate of mortality than men [25] that has been attributed to differences in sex hormone levels [26]. At present, the available cardiovascular outcome data in TG men show that T treatments do not induce increased adverse cardiovascular outcomes. Indeed, most TG men are still relatively young at the moment, when the risk of cardiovascular events is low [8]; in this respect, long-term data and data from older TG men are needed. In TG women, a recent systematic review and meta-analysis did not find an increased risk of myocardial infarction, stroke, or venous thrombosis. Thrombosis risk in TG women may be increased due to the known prothrombotic actions of oestrogen [15]. It is known that oestrogen has strong influences on the sexually dimorphic baseline physiology of the heart and on different cardiovascular diseases. It should be considered that several EDs are oestrogenic and they have been identified as risk factors contributing to the onset of hypertension [27]. For example, some epidemiological studies reported a positive correlation between elevated levels of urinary bisphenol A (BPA) and hypertension [28,29,30]; organochlorine pesticides, phthalate esters and polychlorinated biphenyls (PCBs) exposure may be positively correlated with cardiovascular disease incidence [31]. Overall, considering the current data, including the impact of EDs on cardiovascular health, it is of pivotal interest to explore the potential effects of ED and HT co-exposure to actively preserve TG people’s health.

### 3.2. Carcinogenicity

Cancer risk may be altered in TG people receiving HT, above all considering that the incidence of some types of cancer differs between men and women and some of these differences are attributed to sex hormones [32]. At present, there are not sufficient data to determine the long-term effects of gender-affirming HT on TG individuals’ cancer risk [33]. Concerns around carcinogenicity of long-term HT have been expressed for TG men, although these concerns are not supported by the available data [15]. However, the lack of cancer data underlines the need for studies with large and inclusive sample sizes and long-term follow-up from multiple centres. The prevalence of hormone-sensitive cancers, such as breast and prostate cancer, appears to be low among TG women; however, recent data point out an increased risk of breast cancer and endocrine gland cancers in TG women compared with matched cisgender women, whereas the risk of prostate cancer is decreased compared with matched cisgender men. Other studies reported a low risk of prostate cancer in TG women [15]. ED exposure can also promote the onset and progression of cancer and the link between ED exposure and cancer has already been established for several EDs which are clearly carcinogens [34,35]. As an example, dioxins, Bis (2-ethylhexyl) phthalate and BPA can bind and stimulate oestrogen receptors (ERs), thus contributing to the development of oestrogen-dependent cancers (prostate and breast) [36]. A study conducted to evaluate the relationship between circulating serum EDs and cancer indicated that serum concentrations of BPA and mono-ethyl phthalate were positively correlated with an indicator of breast cancer risk (mammographic breast density) [37]. In another study, phytoestrogens, PCB and dioxins are linked to the development of breast cancer, while arsenic and PCB were reported to significantly contribute to the incidence of prostate cancer [38]. Further studies are needed to clarify the relationship between gender-affirming HT and cancer risk and the possible synergic effects of ED exposure.

### 3.3. Metabolic System

To date, few studies have been performed to evaluate the changes in body composition and body fat distribution in TG individuals. In general, gender-affirming HT on TG women induces a decrease in lean body mass and an increase in fat mass; conversely, HT on TG men produces an increase in lean body mass and a decrease in fat mass [39]. Moreover, TG men showed decreased low-density lipoprotein (LDL) cholesterol and sex-hormone binding globulin and increased triglyceride serum levels. TG women experience increased LDL-cholesterol, triglyceride and sex-hormone binding globulin serum levels [15]. Several studies reported that the regulation of energy metabolism, adipose tissue structure and the control of appetite and satiety may be affected by EDs called “obesogens” [40]. Compounds to which the human population is exposed on daily basis, e.g., phthalates, BPA, PCBs and dioxins, can also promote fat/lipid biosynthesis and storage (weight gain) by compromising energy homeostasis and the basal metabolic rate. Obesogenic EDs act by interfering with nuclear transcriptional regulators that control lipid flux and/or adipocyte proliferation/differentiation, especially the peroxisome proliferator-activated receptor family and steroid hormone receptors [41,42]. It is interesting to notice that the structures of obesogenic EDs are mainly lipophilic and they have the ability to bioaccumulate in the adipose tissue, resulting in the exacerbation of their endocrine disrupting effects; since gender-affirming HT changes the body composition, as a consequence, it might modify the retention of lipophilic pollutants with a possible adverse action not yet studied. Obesity can also influence other diseases, such as female and male infertility, through various mechanisms [43,44]. Recent data showed that TG men undergoing HT may be at risk for insulin resistance; in fact, T administration to females suppresses circulating insulin levels, up-regulates components of the insulin-signalling pathway in liver and suppresses insulin signalling in white adipose tissue [45]. Moreover, several studies have demonstrated clear differences in the prevalence of metabolic disorders in males and females, but the biological basis of the dissimilarities remains to be elucidated, focusing on the multifunctional role of liver in mammals. Liver is a central metabolic organ playing a pivotal role in metabolic processes regulated by steroid hormones and a bidirectional link exists between sex hormones and liver activity, since both oestrogen and ARs are present in the liver in both sexes. The ERα expressed in the liver has a direct effect on the regulation of the hepatic genes relevant for energy and drug metabolism. The sex-related differential expression of the hepatic ERα raises the questions as to whether this receptor is responsible for the sexual differences observed in the physiopathology of the liver [46]. In light of this, an in-depth understanding of the sex-specific mechanisms driving xenobiotic metabolism is of paramount importance for the comprehension of the molecular bases of a large number of metabolic disorders [47] and, in particular, to understand the potential interaction between HTs and obesogenic EDs as well as the impact on TG people metabolism.

### 3.4. Bone Health

Bone is a complex, multifaceted tissue involved in mobility and mineral metabolism; moreover, it is the mesenchymal or stromal and hematopoietic progenitor or stem-cell breeding organ, and the sex steroid hormones play important roles in bone growth and maintenance [48]. Prior to T therapy, TG men have similar bone mass density (BMD) to cisgender females and HT preserves bone density due to the aromatization of T to E2. There are very limited data on the risk of osteoporotic fractures in TG men; however, in TG men who underwent ovariectomy, bone loss has been described when they irregularly used or stopped androgen therapy or when dosage was inadequate. The fracture rate associated with HT in TG women is unknown. Oestrogen is critically important for preserving BMD in post-menopausal women and in men who lack oestrogen action at the bone (e.g., mutations in the ERs or aromatase enzyme). The reasons why some TG women show lower bone density than expected for their age are not clear; it has been hypothesized that there is a link with the decreased outdoor physical activity that may induce a low vitamin D status [15]. To date, an endocrine role has been attributed to bone due to its ability to produce at least two hormones (osteocalcin and fibroblast growth factor 23) and to participate directly or indirectly in leptin, insulin, oestrogens and serotonin signalling. Besides, bone is a well-known oestrogen-responsive tissue reacting to environmental influences [48]. Thus, EDs can exert adverse effects on bone tissue through the alteration of bone modelling and remodelling via changes in bone paracrine hormone synthesis, in the release of systemic hormones, cytokines, chemokines and growth factors, affecting stem cell fate, as well as bone marrow mesenchymal stem-cell differentiation. Recent studies indicated that EDs such as perfluoroalkyl substances, phthalates, BPA, alkylphenols, dioxin and dioxin-like compounds have bone-disrupting effects [49]. In addition, in this case, it appears of pivotal relevance to evaluate the impact of ED exposure on TG people under HT focusing on the bone system.

### 3.5. Reproduction and Fertility

Before starting HT, TG people should be aware that crucial modifications will occur in their reproductive systems, including fertility [15]. In general, HT has a partially reversible impact on fertility, whereas surgical intervention in TG men results in an irreversible loss of natural reproductive capacities. Considering TG women, since HT induces sperm production decline, it has been recommended to discuss fertility options prior to the initiation of HT. The cryopreservation of sperm, oocytes, embryos and—as a more innovative technique—ovarian tissue can be suggested as a fertility preservation option, but it is scarcely used, since it is very expensive. Scarce data are available on the time and manner needed to recover physiological reproductive functions after HT and on the potential detrimental effects of HT [15]. It is known that oestrogen/androgen pathways are of utmost importance in gonadal development, determination of secondary sex characteristics and gametogenesis. EDs affect pathways leading to developmental and reproductive abnormalities in humans and animals; in fact, they can bind the receptors affecting the normal functioning of the pathway, acting either as agonists or antagonists. Almost all major categories of EDs studied were found to generate reproductive problems through the direct binding to either ARs and/or ERs or the activation of alternative receptors such as G protein-coupled receptors (GPCR), GPCR30 and oestrogen-related receptor (ERRγ) [50]. As mentioned above, ED exposure can also affect fertility via indirect mechanisms involving metabolic pathways mainly mediated by hepatic function [51]. More than in other body districts, the HT and EDs mode of actions merge in the reproductive functions of TG people; in this respect, the development of targeted animal models can represent a reliable tool to evaluate the effects and to study appropriate measures to preserve fertility.

### 3.6. Vulnerability Linked to Modified Behaviour (Lifestyles, Food Habits)

Beyond the potential enhanced susceptibility of TG people due to HTs, there would be other factors needing attention, e.g., specific vulnerability. Indeed, the vulnerability of people derives from social, environmental, political, cultural and economic contexts. In this sense, vulnerable groups are not only at risk because they are exposed to a hazard but also as a result of everyday patterns of social interaction and organization, as well as the access to resources [52].

People and/or communities are differentially exposed and vulnerable and this is based on the health status, including disability linked to specific diseases, gender, age, wealth, education, race/ethnicity/religion and class/caste. The lack of resilience and the capacity to cope with and adapt to extremes and changes are important causal factors of vulnerability. In general, health disparities for TG adults are well documented, but health data are lacking and this makes it more difficult to set appropriate interventions [53,54].

The health effects of food insecurity—defined as the lack of “consistent, dependable access to adequate food for active, healthy living”—include, but are not limited to, obesity, diabetes, hypertension, depression and anxiety. Food insecurity is associated with several factors and TG communities have also been identified as being at elevated risk [55]. If the vulnerability of TG people linked to cultural, psychological and social factors is known, other specific vulnerabilities have been recorded.

As an example, TG populations are considered as a highly vulnerable group to HIV infection [56,57]. Moreover, TG people have shown to be a specific target of marketing strategies of tobacco companies which use price discounts, including coupons and rebates, to market their products; in this sense, it contributes to high tobacco use among TG people and the related consequences on health [58]. Reducing harmful oral hygiene practices can minimize the negative impacts of periodontal diseases and TGs are among the most vulnerable groups [59].

Concerning the use of cosmetics—higher in TG people in comparison to cisgenders—the adverse effects of the fragrance constituents, such as phthalates, paraben, glutaraldehyde, hydroperoxides, oil of turpentine, metals, nitro musks and essential oils, among others, are being identified. The endocrine–immune–neural axis perturbation pathways of these chemicals have been proved [60]. Moreover, the sc injections, or “fillers,” are used illicitly and in large quantities by TG women for feminization. They are associated with severe complications, but data on their use are limited [61].

## 4. Rodent Models

### 4.1. State of the Art

Considering HT for the female-to-male (FtM) transition, few studies focused on the use of the animal models mimicking gender-affirming HT are available. Two studies investigated the effect of HT on bone health and atherosclerosis in ovariectomized adult female mice. In both studies, mice received 31 µg/week of T by sc injections and the treatment started at 6 or 10 weeks and continuing through 20/22 weeks of the mice’s age. The early groups (6 weeks) mimicked the adolescent transition and the late groups (10 weeks) the transitioning during early adulthood [62,63]. Although these studies did not investigate reproductive changes, at the end of the treatment, the authors verified that the T serum levels were significantly increased in respect to the control groups, without effects on E2 serum levels in the first study [62] or increased levels in the second study [63]. This approach mimics gender-affirming HT in times and route of administration, but, considering the results on E2 serum levels, it has been postulated that the dose of T used in this study was too low to adequately reproduce HT. Another study evaluated the effects of T on renal morphology and function using a rat model of T therapy. Female rats were im treated with T (3.0 mg/kg body weight) every 10 days for 12 weeks. At the end of treatment, T serum levels were increased at the cisgender concentrations and the findings suggested a renal adaptation and an increased lean mass which resembled that observed in TG people exposed to long-term T therapy [64]. A recent study proposed a mouse model to study the reproductive effects of gender-affirming HT by administering, twice a week, T sc injections at a 0.225, 0.45, or 0.90 mg/dose to adult female mice (8–9 weeks of age) for 6 weeks. The study analysed different reproductive endpoints, namely, organ weight, vaginal cytology, serum level of T, LH, FSH and Anti-Müllerian hormone, and ovary histology and morphometric analysis of follicle distribution were conducted. On the basis of the results, it was established that 0.45 mg per dose, twice a week, was ideal to mimic HT in the mouse model since mice exhibited several reproductive perturbations also observed in TG men, such as acyclicity, T in male range, LH reduction and ovarian phenotype similar to polycystic ovary morphology with complete absence of corpora lutea. The persistence of these changes after cessation of HT was not evaluated, although the results suggested that the ovarian reserve was not depleted by T treatment [65]. In another study, mice (6 weeks old) were injected sc with 0.40 mg of T once a week for 6 weeks and were sacrificed 1 or 6–7 weeks after the last administration. T therapy induced signs of virilization as clitoromegaly and cessation of oestrus cycles. Furthermore, one week after the last treatment, the T serum levels were increased, with all values within the normal range reported in adult male mice, and the ovaries were smaller than control mice but contained a similar number of follicles and a significant reduction in corpora lutea number. After 5 weeks without HT, the T serum levels declined to baseline level, they returned cycling, clitoromegaly was no longer apparent and only the ovary weight remained lower than that of control mice [66]. Substantially, the effects on reproductive functions seemed to be reversed after the cessation of T therapy. However, the presence of scarce corpora lutea in T-treated mice denoted that the oestrus cycle was inhibited. In this study, the time of exposure and the T dose level induced phenotypic characteristics in mice such as in TG men, but they did not completely mimic the human situation, in that HT lasts several years and TG men have cessation of menses [15]. A second recent study showed the reversibility of acyclicity in female mice subcutaneously implanted with T (5 mg/pellet, 60-day release) or placebo pellets for 6 weeks. After four cycles following T pellet removal, no differences were detected in the key reproductive parameters of T treated and control mice [67].

Considering the feminization HT for male-to-female (MtF) transition, to date, only one in vivo animal study intentionally mimicking gender-affirming HT is available and the principal objective was to evaluate the morphology and metabolomic profile of the brain. Male rats (8.5 weeks old) received daily sc injections of E2 (0.2 mg/kg) or E2 plus CPA (0.2 mg/kg + 0.8 mg/kg) or 1.2 mg/kg propanodiol for 30 days. The CPA dose was equivalent and the E2 dose was slightly higher than that prescribed for TG women (70 kg) under HT. Both E2 and E2 + CPA decreased the level of plasma T and testicular weight [68]. Although many limitations are present, such as a confuse picture of the treatment and short-term administration, for the first time, an MtF rat model was used to reproduce some results observed in human adult male undergoing HT.

### 4.2. New Perspectives

In light of recent research, it can be sustained that animal models can provide a tool to better understand the impact of gender-affirming HT on the health of TG men and women.

Among the animal species currently used in the hazard identification, probably non-human primates represent the best choice, but they have many drawbacks; in particular, beyond the recognized ethical issues, they are cost-prohibitive and have a long reproductive life span and gestational cycle [69]. Sheep models have analogous limitations [70]. The use of rodent models—as for other contexts—to mimic HT therapy in TG men and women shows limitations and they may not perfectly match human physiology. Nevertheless, rodents have the great advantages that they are easy to handle and maintain, are relatively inexpensive, have a well-characterized reproductive cycle, a short gestational cycle and an accelerated life span [69]. Moreover, considering the RA of chemical contaminants and related effects, huge databases of historical data are available for rodent models. Indeed, the Organisation for Economic Co-operation and Development Test Guidelines, developed to study the health hazard likely to arise from exposure to chemicals, are designed for use with rodents and rats are the best choice to study the ED effects and actions [71,72]. In this respect, the development of dedicated animal models that mimic gender-affirming HT can be based on rodents—and rats in particular—considered suitable to verify the HT’s long-term effects and the potential specific susceptibility and vulnerability of TG people.

A crucial factor in the study of gender-affirming HT is the duration of treatment and the age correlation between humans and animals (Figure 2). HT can commonly start during adolescence (Tanner stages 2 to 3) [15], corresponding to 8–9 weeks in rodents [73], and it lasts for a long time, potentially even for the whole life of TG people. When a research study focuses on reproductive and fertility effects, the minimum time of HT administration in rat should be 90 days, corresponding to 8–9 years in humans [73], to allow a congruous recovery time without HT to revert cyclicity in females, while remaining in the time window to possibly become pregnant, and spermatogenesis in males. The age correlation and the duration of treatment are also important for the evaluation of potential long-term effects of HT on the different body districts and they can be modulated accordingly.

Concerning the parameters to be evaluated in the model, as already described, the available animal studies mimicking the gender transition reported, as a minimal criterium, the measurement of T serum concentration at the end of HT. In fact, the T serum level in the cisgender range is considered the best biomarker to evaluate the success of gender-affirming HT for both TG men and women [15,21]. The value of the rodent model also relies on the fact that other endpoints can be evaluated in order to obtain the confirmation of HT’s efficacy, with the final aim of setting a panel of biomarkers to be used to better identify and characterize the MtF or FtM models. In fact, the specific parameters which unambiguously define the model can be used as starting point on which further studies concerning the reproductive/fertility capacity, the exposure to chemicals, the impact of lifestyles, etc. can be based. Such parameters can be divided into two groups, namely, functional biomarkers to be evaluated in living rodents and tissue biomarkers after sacrifice. Among functional biomarkers, beyond T analysis, E2 serum levels as well as hypothalamic releasing factors, the pituitary hormones and Anti-Mullerian hormone are suggested, accompanied by the evaluation of the oestrus cycle in female rodents and sperm analysis in males, particularly relevant also in consideration of fertility and reproductive issues. Specific markers, e.g., the gonadal peptide inhibin B, biomarker of the seminiferous epithelium status and predictor of ovarian reserve and cytochrome P450 aromatase, the enzyme that irreversibly catalyses the conversion of androgens into oestrogens, can be included to better characterize the TG status.

Beside the above-mentioned parameters, specific functional and tissue biomarkers can be selected according to the aim and targets of the study to be performed, e.g., focused on bone health, cardiovascular system, carcinogenicity, etc. Moreover, the selection of appropriate parameters should take into account the different incidence—if recorded—of alterations/disease and tumours according to sex, comparing the incidence of the birth-assigned sex with the incidence of sex of destination.

In addition, to evaluate the impact of chemical contaminants, several toxicological parameters linked to known mode of action of the EDs can be evaluated, e.g., for compounds targeting thyroid homeostasis (as an example, brominated flame retardants, ethylene bisdithiocarbamate fungicide), the measurement of thyroid hormones and thyroid-stimulating hormones is recommended [74].

It is important to remind that, when the targets of HT and ED overlap (e.g., compounds directly interacting with ER and/or AR), the comparison with the negative control in an experimental study will allow researchers to discriminate between them.

As tissue biomarkers, organ weight and histopathological evaluation of reproductive tissues are of pivotal interest to clearly identify their status at different time points from the beginning of HT and, potentially, to set a consistent and reliable recovery period after HT interruption prior to conceiving. In addition, behavioural factors can be monitored in rodents in the absence of sexual dimorphism as indicator of sex transition.

As the last consideration, the integration of phenotypic biomarkers with the recently developed “omics” techniques, namely, genomics, transcriptomics, epigenomics, proteomics, metabolomics and miRNAomics, will allow researchers to evaluate, more comprehensively, several aspects of the efficacy but also of the long-lasting and unexpected side effects of gender-affirming HT, including cellular mechanisms in physiological conditions or following chemical exposure [75].

## 5. Discussion

TG people are known to experience a high prevalence of adverse health outcomes, including HIV and other sexually transmitted infections, mental health distress and substance use and abuse [76]. In fact, TG people, as other gender and sexual minority individuals, meet significant health disparities in terms of risk factors, health outcomes and access to health care; the latter appears to be both the direct and indirect consequence of oppression, prejudice and discrimination. Indeed, the interplay between the stress of concealment or self-shame, discrimination or victimization and poor health outcomes leads to increased psychological distress and suicidality [77]. Moreover, gender-affirming HT—usually lasting for a long time, often for all the person’s life—poses the questions as to whether it can induce persistent alterations or enhance the incidence of specific diseases (e.g., endocrine, metabolic, etc.). In addition, similar to the cisgender population, TG people are continuously exposed to environmental and food contaminants, including EDs, the effects of which might be exacerbated by HT. Indeed, since in the last years TG people are becoming more visible in the society, maybe due to significant socio-cultural changes, biomonitoring as well as longitudinal studies specifically addressing TG people should be performed in order to provide scientific and up-to-date epidemiological data.

In this context, in order to better preserve TG people’s health and to limit the disparities, the development of an animal model should be considered appropriate, as it can represent an important tool to evaluate the impact of gender-affirming HT, as well as the potential interaction with chemical contaminants or other specific factors of vulnerability.

## 6. Conclusions

The development of an animal model specifically addressing TG people’s health can be useful to generate unique datasets to support RA in TG people that—at present—cannot be obtained for the lack of specific assays. Subsequent studies would contribute to a better level of protection for TG people, a population group who can be identified as highly susceptible to specific chemical hazards. In conclusion, since the scarce available scientific evidence indicates the need for toxicological testing addressing the pharmacological gender-affirming transition, the development of targeted experimental tools appears as the necessary step for a more robust hazard identification in TG women and men, leading to a more reliable RA.

## Figures and Tables

**Figure 1 ijerph-18-12640-f001:**
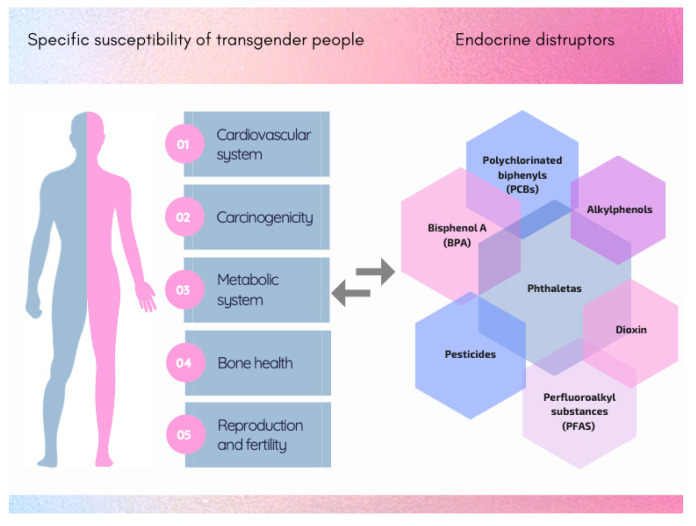
Schematic representation of the major systems involved in specific susceptibility of transgender men and women and its possible relationship with endocrine disruptors.

**Figure 2 ijerph-18-12640-f002:**
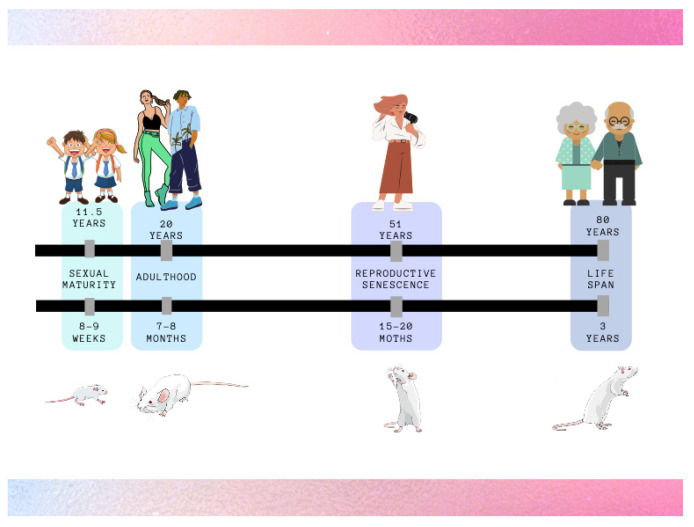
Comparative timelines between human and rat.

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
