# Peer review of "Rodent Model of Gender-Affirming Hormone Therapies as Specific Tool for Identifying Susceptibility and Vulnerability of Transgender People and Future Applications for Risk Assessment"

_ijerph, 2021, doi:10.3390/ijerph182312640_

Round 1
Reviewer 1 Report
The manuscript reviewed the rodent models of gender affirming hormone therapies. The paper adds to our current understanding on the transgenders and their hormone therapy. There are a few major comments for their consideration.
Comment 1. The author should give a more detailed definition on the transgender, what are their main physical characteristics, is it mostly a result of nature or environment?
Comment 2 line 37-40, the author mentioned that overall prevalence for transsexualism with an increase during the last 50 years, what causes this phenomenon, and is it related to environmental endocrine disruptor?
Comment 3 line 63, what is the full name of qof ?
Comment 4 line 187-189 Whether endocrine disruptors can cause abnormalities in the cardiovascular system of transgender people, and whether there are relevant epidemiological studies?
Comment 5 line 209-219, 219-257, 259-282, the question is similar to the above one, did these endocrine disruptors have cancer risks, effect metabolic system or bone health on transgender people?
Comments 6. Line 284-305, What about the reproduction and fertility on the transgender people, does the HT treatment could improve their reproductive status or cause new reproductive problems?
Author Response
Thank you for the constructive review, the referee’s comments have improved our manuscript.
The manuscript reviewed the rodent models of gender affirming hormone therapies. The paper adds to our current understanding on the transgenders and their hormone therapy. There are a few major comments for their consideration.
Comment 1. The author should give a more detailed definition on the transgender, what are their main physical characteristics, is it mostly a result of nature or environment?
ANSWER 1. Thank you for the comment; although the information requested by the Referee are already present, the introduction has been improved as suggested
Comment 2 line 37-40, the author mentioned that overall prevalence for transsexualism with an increase during the last 50 years, what causes this phenomenon, and is it related to environmental endocrine disruptor?
ANSWER 2. The reasons behind the increase of TG people and transsexualism have been deepened and the potential impact of ED exposure has been discussed
Comment 3 line 63, what is the full name of qof ?
ANSWER 3. Thanks for the comment, the abbreviation ‘qof’ was deleted from the manuscript.
Comment 4 line 187-189 Whether endocrine disruptors can cause abnormalities in the cardiovascular system of transgender people, and whether there are relevant epidemiological studies?
Comment 5 line 209-219, 219-257, 259-282, the question is similar to the above one, did these endocrine disruptors have cancer risks, effect metabolic system or bone health on transgender people?
ANSWER 4 and 5. To our knowledge, there are no specific epidemiological studies that dealing with effects of endocrine disruptors in TG people. Indeed, no specific data are present on ED alterations in the cardiovascular, metabolic or bone systems as well as no association with cancer risks in TG people. Biomonitoring studies on the exposure to endocrine disruptors are completely lacking for transgender people.
Comments 6. Line 284-305, What about the reproduction and fertility on the transgender people, does the HT treatment could improve their reproductive status or cause new reproductive problems?
ANSWER 6. On the basis of the scarce available literature data, and on the basis of the known mechanism of action of gender affirming HT, it is plausible that HT has detrimental effects on reproductive health of TG people; in fact, the cryopreservation of sperm, oocytes and ovary tissue is often suggested before the HT. Indeed, such aspects linked to (long-lasting) effects of HT are among the main aims of the animal model development
Reviewer 2 Report
Thank you to give me the possibility to improve and read this research paper.
The text needs some minor revisions.
The authors should point out that the health risk of transgender people is only one of the experiences of this subgroup. It must be emphasized in the introduction and conclusions that the identity and life of trangender and transsexual people is not limited to taking drugs and that looking at health aspects is only one point of view. The article should avoid medicalizing trans people.
It is useful to insert the age of trans people in relation to the outcomes of hormone intake and to indicate the age groups at risk (see Scandurra et al.) . it is also useful to point out the need for longitudinal studies in the conclusions and to include monitoring policies.
Scandurra, C., Carbone, A., Baiocco, R., Mezzalira, S., Maldonato, N. M., & Bochicchio, V. (2021). Gender Identity Milestones, Minority Stress and Mental Health in Three Generational Cohorts of Italian Binary and Nonbinary Transgender People. International Journal of Environmental Research and Public Health, 18(17), 9057.
Author Response
Thank you for the constructive review, the referee’s comments have improved our manuscript.
Thank you to give me the possibility to improve and read this research paper.
The text needs some minor revisions.
The authors should point out that the health risk of transgender people is only one of the experiences of this subgroup. It must be emphasized in the introduction and conclusions that the identity and life of trangender and transsexual people is not limited to taking drugs and that looking at health aspects is only one point of view. The article should avoid medicalizing trans people.
ANSWER. Thank you for the interesting consideration. As described in the text, it should be kept in mind that focus of the paper is the risk assessment and the development of a targeted animal model for toxicological evaluation of the effects of chemical exposure as well as of potential long-lasting side effects of HT. The Authors didn’t intend to medicalize TG people; on the contrary, the Authors are aware that – in the frame of risk assessment - the TG people represent a sub-group of population with specific vulnerability and susceptibility linked to – among other factors – HT. For this reason, as for children, elderly or pregnant women (recognized as vulnerable and susceptible sub-groups of population in risk assessment), they need dedicated tools to ensure a more reliable and specific health risk assessment.
It is useful to insert the age of trans people in relation to the outcomes of hormone intake and to indicate the age groups at risk (see Scandurra et al.), it is also useful to point out the need for longitudinal studies in the conclusions and to include monitoring policies.
ANSWER.The above mentioned suggestions have been taken into consideration
Scandurra, C., Carbone, A., Baiocco, R., Mezzalira, S., Maldonato, N. M., & Bochicchio, V. (2021). Gender Identity Milestones, Minority Stress and Mental Health in Three Generational Cohorts of Italian Binary and Nonbinary Transgender People. International Journal of Environmental Research and Public Health, 18(17), 9057.
ANSWER. Thanks for interesting reference, it was inserted in the manuscript
Reviewer 3 Report
The manuscript entitled Rodent models of gender-affirming hormone therapies: state of the art and future applications for risk assessment is a non-systematic literature review. The authors describe the hardships of transgender individuals in daily life due to the hormone therapy's adverse side effects coupled with the surrounding environment. The authors proposed a new risk assessment approach for transgender individuals, using the Rodent models, to generate enough data to understand better the impact of gender-affirming hormone therapies on the health of transgender individuals.
The manuscript was well-written and organized, and the topic is relevant at this point. The authors employed a non-systematic review and primarily based their review on a knowledgeable selection of current, high-quality articles on the topic of interest and did not follow the PRISMA guideline that the Journal requires for critical review type of publication.
The authors need to thoroughly proof the manuscript to correct and revise the spelling, grammar, and word choice.
I do not understand the authors' choice of words "state of the art."
The title did not accurately describe the manuscript's content. The authors delineated the adverse effects of hormone therapy in more than half of the manuscript. Perhaps, the title should also reflect that content.
Author Response
Thank you for the constructive review, the referee’s comments have improved our manuscript.
The manuscript entitled Rodent models of gender-affirming hormone therapies: state of the art and future applications for risk assessment is a non-systematic literature review. The authors describe the hardships of transgender individuals in daily life due to the hormone therapy's adverse side effects coupled with the surrounding environment. The authors proposed a new risk assessment approach for transgender individuals, using the Rodent models, to generate enough data to understand better the impact of gender-affirming hormone therapies on the health of transgender individuals.
The manuscript was well-written and organized, and the topic is relevant at this point. The authors employed a non-systematic review and primarily based their review on a knowledgeable selection of current, high-quality articles on the topic of interest and did not follow the PRISMA guideline that the Journal requires for critical review type of publication.
The authors need to thoroughly proof the manuscript to correct and revise the spelling, grammar, and word choice.
ANSWER. Thank you for the comment; the English revision has been performed
I do not understand the authors' choice of words "state of the art."
The title did not accurately describe the manuscript's content. The authors delineated the adverse effects of hormone therapy in more than half of the manuscript. Perhaps, the title should also reflect that content.
ANSWER. Thank you for your comment. The manuscript emphasizes the specific susceptibility and vulnerability of transgender people due to hormone therapy; indeed, this aspect has been introduced in the title in order to take into account the Reviewer’s suggestion. The title has been changed in “Rodent model of gender affirming hormone therapies as specific tool for identifying susceptibility and vulnerability of transgender people and future applications for risk assessment”